# Spatial Differentiation, Influencing Factors, and Development Paths of Rural Tourism Resources in Guangdong Province

Chenmei Liao [1,†], Yifan Zuo [2,†], Rob Law [3], Yingying Wang [2] and Mu Zhang [1,*]

1   Shenzhen Tourism College, Jinan University, Shenzhen 518053, China
2   School of Physical Education, Shenzhen University, Shenzhen 518061, China
3   Asia-Pacific Academy of Economics and Management, Department of Integrated Resort and Tourism Management, Faculty of Business Administration, University of Macau, Macau 999078, China
*   Correspondence: zhangmu@jnu.edu.cn; Tel.: +86-755-2693-1865
†   These authors contributed equally to this work.

**Abstract:** Rural tourism resources are the core carriers of rural tourism. It is, therefore, beneficial to further optimize the layout of rural tourism and to explore the spatial differentiation of rural tourism resources and their influencing factors. Taking 4670 rural tourism resources in Guangdong Province in China as the research object, this study explores the spatial distribution patterns of rural tourism resources through the nearest neighbor index, grid dimension analysis, kernel density analysis, and standard deviation ellipse method. Geodetectors are used to identify the influencing factors of the spatial heterogeneity of these resources in Guangdong Province. The findings reveal the following: (1) The distribution of rural tourism resources in Guangdong Province shows a tendency of agglomeration along the Tropic of Cancer, and the spatial distribution is unbalanced. The hot and cold spots show a "northeast-southwest" distribution trend. Furthermore, most of the hotspots form three high-density core areas, the sub-dense stretch zones connect into a w-shaped belt, and the sub-cold areas and sub-hot areas show a large expansion trend, thus forming five radiation areas. (2) The distribution of rural tourism resources in Guangdong Province is affected by multiple factors. In particular, the force of agricultural resource endowment, tourism resource endowment and transportation location are relatively strong, and social economy and tourist source market are the weak factors.

**Keywords:** rural tourism resources; spatial distribution; geodetectors; development paths

## 1. Introduction

Rural tourism—a multi-industry sector that promotes the economic, social, and spatial transformation and reconstruction of rural areas—has become a widely used means of rural regeneration and conservation in recent years [1,2]. Since the 1980s, people have paid more attention to the ecological environment, so the worldwide "green movement" has begun to promote the development of rural tourism and gradually made it an important component of sustainable tourism in developed countries. In the 1990s, under the vigorous promotion of the World Tourism Organization, rural tourism as a form of ecotourism was promoted in developing countries. In some countries, rural tourism is considered an effective means of preventing agricultural decline and increasing rural incomes [3]. In the United States, 30 states have tourism policies specifically targeting rural areas, and 14 of them include rural tourism in their overall tourism development plans. Travel pattern data in western North Carolina was analyzed using GIS to develop rural tourism potential marketing [4]. In Israel, the development of rural tourism is used as an effective supplement to the decline of rural income, and the number of rural tourism enterprises is increasing annually. As such, the concept of service quality and service orientation in rural tourism establishments in Israel was investigated [5]. Other countries (e.g., Canada, Australia, and New Zealand) and regions (e.g., the former Eastern Europe and the Pacific)

consider rural tourism as a driver of economic development and economic diversification in rural areas [6,7]. Some authors attempted to investigate the extent to which regional development can be brought forward by tourism in rural regions and what can be learned from the model project for shaping similar support activities in the future. Furthermore, other countries have created several successful rural tourism products based on their own characteristics, and there are lessons to be learned from their experiences. For example, in the development of green tourism products, Japan pays attention to environmental protection and local community power [8]. In another case, managers of agrotourism areas in Italy make use of the rich natural resources unique to the countryside to turn these resources into living spaces with multiple functions, such as education, recreation [9], and culture [10]. France, with its vast arable land and rich agricultural resources, has chosen the traditional tourism development model in the construction of its rural leisure agricultural tourism, maintaining the authenticity and local nature of tourism products, and devoting itself to the development of traditional agricultural production [11]. As a result, the French rural tourism sector provides tourists with a refreshing experience close to tradition and nature, while highlighting the uniqueness of French rural tourism. The achievement of Polish agritourism is the creation of a new form of recreation for tourists, permitting accommodation on farms and farm visits [12]. The Polish practical experience of agritourism is regarded as positive.

The official website of the United Nations World Tourism Organization launched the 2022 "Best Tourism Village" selection campaign, which aimed to promote rural culture and natural resource protection as well as sustainable social and economic development through tourism. Since the 1980s, rural tourism in China has basically completed the initial stage and entered a period of rapid development. In particular, thanks to the construction of ambitious high-speed railways, highways, and rural roads, as well as the proliferation of private cars, China has created an unprecedented rural tourism market [13]. The Guangdong Provincial Department of Culture and Tourism announced that in the first half of 2018, the province received 337 million rural tourists, accounting for 56.25% of the total tourists in the country, comprising 30.3% of the total rural tourism income. Indeed, rural tourism has become an important part of the tourism industry in Guangdong Province. Therefore, to promote its sustainable development, Guangdong Province released the first batch of rural tourism development resources catalog in June 2019, which integrated the resources of 21 cities. The development of rural tourism resources is vital in inheriting rural culture, improving the public service system, and promoting industrial integration. One study has concluded that the characteristic of its spatial layout will directly affect the rationality of the spatial allocation of tourism resource elements, studying the factors influencing spatial pattern can expand the ideas for improving the governance of tourism regional spatial units, and the scientific development path promotes the promotion of efficient development of rural tourism [14]. Therefore, exploring the spatial differentiation patterns of rural tourism resources and their influencing factors is conducive to further optimizing the rural tourism layout, thus providing reference values for promoting rural tourism in developing countries around the world.

Many scholars have explored issues related to rural tourism from different perspectives. Existing studies have mainly tackled conceptual frameworks, consumer markets, and development paths. Thus, a more complete research system has been formed, with more meaningful results in industrial development and model exploration, resource evaluation and development, spatial distribution, and various influencing factors. For instance, Frisvoll discussed the fusion relationship between authenticity and rural tourism [15], while Lane and Kastenholz studied the development and change in rural tourism from the theme category and geographical distribution [2]. Zhou identified the attributes of the destination image and linked it to the concept of rural and tourism imagination, concluding that the image of rurality in tourism imagery is positive and market-oriented [16]. Differences in tourism resources are the prerequisites and material bases for rural tourism development, and rural tourism resources in different regions lead to variations in rural

tourism development strategies [17]. Wang characterized rural tourism resources through a method of character selection to sort out rural development paths [18]. Mitchell and Shannon found that the spatial distribution characteristics of rural tourism resources tend to influence the behaviors of tourists and the layout of the industry [19].

At present, only a few studies have focused on the spatial structure or distribution of rural tourism and related influencing factors. For example, Sang-Hyun Lee et al. demonstrated the applicability of centrality indices for evaluating spatial characteristics using geographic information systems (GIS) and network analysis [20]. Xue et al. explored the network degree and development trends in rural tourism resources, while Kumar et al. analyzed the interrelationship between factors conducive to rural tourism development in India [21]. Jiang et al. evaluated the resource control ability of rural tourism networks by determining the resource control relationship and assessing the structure of the rural tourism network [22]. Some authors also studied the characteristic features of health tourism destinations from a regional and spatial perspective [23]. Nevertheless, academic circles have not sufficiently probed the issues of rural tourism and mainly focused on account of market, economy, or management.

Meanwhile, Guangdong Province is located in the southern part of China and has a complex and diverse landscape with mountains, hills, and plateaus and plains, which account for 33.7%, 24.9%, 14.2%, and 21.7% of the province's total land area, respectively, while rivers and lakes account for only 5.5% of the province's total land area. As of 31 December 2021, there are 21 prefecture-level cities in the province. The villages of Guangdong Province are rich in natural tourism resources and are considered to have great potential for rural tourism. Therefore, this paper takes the resources listed in the catalog of rural tourism development resources released by Guangdong Province as the research samples and uses the nearest point, geographic concentration, and imbalance indexes, grid-dimensional analysis, spatial autocorrelation analysis, kernel density estimation, standard deviation ellipse analysis, and geographic probe analysis. The goals of this study are to investigate the spatial distribution characteristics and influencing factors of the resources, to provide a quantitative basis for optimizing the integration of rural tourism resources in developing countries, and to seek a reasonable rural tourism development path based on these results.

## 2. Materials and Methods

### 2.1. Methods

2.1.1. Average Nearest Neighbor, Geographic Concentration Index, and Disequilibrium Index

The nearest neighbor index reflects the spatial aggregation characteristics of rural tourism resources, i.e., the ratio of the actual to the theoretical nearest neighbor distances [24]. Zuo et al. used the nearest neighbor distance to measure the mutual proximity of sports tourism resources in the spatial distribution [25] This is computed using the following formula:

$$r_I = \frac{r_O}{r_E} = 2\sqrt{D} = 2\sqrt{\frac{m}{A}} \tag{1}$$

where $r_I$ represents the nearest neighbor index, $r_O$ represents the average nearest neighbor distance, $r_E$ represents the theoretical nearest neighbor distance, $D$ represents the nearest neighbor density, $A$ is the area of the region, and $m$ is the number of rural tourism resources in Guangdong. When $r_I < 1$, the distribution is agglomerative; when $r_I = 1$, it is random; and when $r_I > 1$, it tends to be uniform. The z-score and $p$-value determine whether it is statistically significant.

Geographic concentration index is an important index that is used to measure the concentration degree of geographical matters [26]. It is calculated using the following formula:

$$G = 100 \times \sqrt{\sum_{i=1}^{n} \left( \frac{X_i}{T} \right)^2} \qquad (2)$$

where $G$ is the geographical concentration index of rural tourism resources in Guangdong, $X_i$ is the number of rural tourism resources in the prefecture-level city, $T$ is the total number of rural tourism resources in the province, and $n$ is the number of prefecture-level cities. The larger the $G$ value, the more concentrated the distribution of rural tourism resources in the province.

The disequilibrium index is used to measure the overall distribution degree [27]. Using the Gini coefficient to calculate the concentration index, the imbalance index $S$ of rural tourism resources in Guangdong can be determined through the following formula:

$$E = \frac{-\sum_1^N P_i \ln P_i}{\ln N} \qquad (3)$$

where $E$ is the Gini coefficient of rural tourism resources in Guangdong, $P_i$ is the percentage of rural tourism resources in each prefecture-level city, and $N$ is the number of prefecture-level cities. The larger the $E$ value, the higher the degree of spatial agglomeration of rural tourism resources in Guangdong Province.

### 2.1.2. Grid Dimension Analysis Method

Grid dimension analysis is an indicator of the complexity and equilibrium of a point set distribution, in which the network dimension N($r$) varies as the network scale X changes when the grid is measured over the spatial extent of the point set [28,29]. When the spatial distribution of the point set is scale-free, the formula is given as follows:

$$\text{N}(r) \propto r^{-T} \qquad (4)$$

where $T = \text{D}_0$ is the capacity dimension. Assuming that the statistical number of grid points is $\text{N}_{ij}$, and the number of points in the full area range is N, the probability can be defined in general terms as $\text{P}_{ij} = \text{N}_{ij}/\text{N}$, and the information dimension formula is given by:

$$I(r) = -\sum_i^k \sum_i^k P_{ij}(r) \ln P_{ij}(r) \qquad (5)$$

In the above formula, K = 1/X is the value of the number of segments of each edge within the region, and if the point set is fractal, the formula is given by:

$$\text{I}(r) = \text{I}_0 - D_1 r \qquad (6)$$

where $D_1$ is the information dimension, and $\text{I}_0$ is a constant, reflecting the degree of equalization of the spatial distribution of the point set. When $D = 2$, it indicates a uniform distribution of regional points. When the value of $D$ tends to 1, this represents the tendency of points to be concentrated into a geographical belt (geographical line). When $D_0 = D_1$, the point set is simply fractal. Both Zuo et al. [25,30] and Qi et al. [31] used Geodetector to study the similarity between the independent and dependent variables in the spatial distribution and understand whether different influencing factors have an interactive effect on spatial distribution.

### 2.1.3. Spatial Autocorrelation (Global Moran's Index)

Spatial autocorrelation reflects the degree of correlation between a certain geographic phenomenon or attribute value on a regional unit and the same phenomenon or attribute

value on an adjacent regional unit [32]. Global spatial autocorrelation is the representation of the global Moran's *I*, which is used to determine whether the spatial distribution characteristics of rural tourism resources in Guangdong are clustered, random, or discrete. The calculation formula of the value of Moran's *I* is as follows:

$$I = \frac{\sum_{i=1}^{n} \sum_{j=1}^{n} \omega_{ij} \left( X_i - \overline{X} \right) \left( X_j - \overline{X} \right)}{S^2 \sum_{i=1}^{n} \sum_{j=1}^{n} \omega_{ij}} \tag{7}$$

where $\omega$ represents the spatial weight between areas *i* and *j*; *n* represents the number of regions; and *Xi* and *Xj* represent the observation values of locations *i* and *j*, respectively. The value range of Moran's *I* is [−1, 1]: Moran's *I* > 0 indicates a positive spatial correlation phenomenon, a value of <0 indicates a negative correlation phenomenon, and a value of 0 indicates an independent random distribution.

### 2.1.4. Kernel Density Estimation Method

The kernel density estimation method can reflect the spatial distribution density and aggregation characteristics of elements. The formula is as follows [33]:

$$Fn(x) = \frac{1}{nh} \sum_{i=1}^{n} \text{T}, \quad \text{T} = k \left( \frac{x - x_i}{h} \right) \tag{8}$$

where *Fn(x)* is the kernel density estimate, *n* is the number of resources, *h* > 0 is the bandwidth, T is the kernel function, and $x - x_i$ is the distance from the estimated point to the measurement point $x_i$. The reasonable spatial search bandwidth *h* of rural tourism resources in Guangdong is set as 50 km by several experiments.

### 2.1.5. Standard Deviation Ellipse Analysis

Standard deviation ellipsometry is a spatial statistical method based on a multi-perspective measure of the spatial distribution and global characteristics of geographical elements and is used for studying the spatial location and spatial structure of an object [34]. The quantitative description of the spatial distribution and evolution of the study object from a global perspective is based on basic parameters, such as centroid, azimuth, long semi-axis and short semi-axis, which can describe the spatial distribution and evolutionary characteristics of a study population [34]. The researcher investigated the distribution of historic villas, palaces and factories using Standard Deviational Ellipse [35]. The standard deviation ellipse analysis is a commonly used method, so the formulas for its measurement are omitted.

### 2.1.6. Geodetector Analysis

Geodetector is a tool used to analyze and detect spatial differentiation by identifying the extent to which a certain factor explains the spatial of the result variable, therefore revealing the source of its spatial difference [36]. The formula is given by:

$$q = 1 - \frac{\sum_{h=1}^{L} N_h \sigma_h^2}{N \sigma^2} \tag{9}$$

where *L* represents the variable stratification, i.e., classification or partition; $N_h$ and N represent the number of units in layer *h* and the entire area, respectively; $\sigma_h^2$ and $\sigma^2$ represent the variance of the result variable in layer *h* and the entire area, respectively; and *q* represents a certain front. The magnitude of influence of the dependent variable on the outcome variable is in the range of [0, 1]. The closer *q* is to 1, the greater the explanatory strength of the pre-dependent variable on the outcome variable. Conversely, the closer *q* is to 0, the smaller the explanatory strength. The current study uses the geographic detector

method to identify the factors affecting the spatial distribution of rural tourism resources in Guangdong.

*2.2. Data Sources*

This study obtained its data from the website of the "Catalog of Rural Tourism Development Resources in Guangdong Province (First Batch)" published by the Guangdong Province Department of Culture and Tourism (http://whly.gd.gov.cn/service_newajjq/content/post_2844888.html accessed on 20 October 2021). Data of 4670 rural tourism development resources from 21 prefecture-level cities in Guangdong Province were extracted, including city, county (city, district), town (street office), administrative village, resource name, basic type, sub-category, main category, and remarks. Two research assistants rearranged the original classification data for statistical purposes. The geographic coordinates of the 4670 rural tourism development resources were picked up one by one using the Baidu Map application call interface, after which they were processed and converted to WGS1984, and then summarized and entered into the GIS system to set up a spatial point database of rural tourism development resources in Guangdong Province. The distribution of rural tourism development resources in the 21 prefecture-level cities of Guangdong is shown in Table 1. In addition, Guangdong vector maps were obtained from the website of the Guangdong Provincial Department of Natural Resources (http://nr.gd.gov.cn/map/bzdt/ accessed on 28 February 2022). Data in the impact factor indicators were obtained from the Guangdong Statistical Bureau's website in the Guangdong Statistical Yearbook 2021 and the Guangdong Rural Statistical Yearbook 2021 (http://stats.gd.gov.cn/gdtjnj/index.htmlaccessed on 18 March 2022). Among these, rural tourism development resources are divided into five main categories: rural industrial integration, rural settlement architecture, rural historical relics, rural folk culture, and rural natural landscape, including 16 sub-categories and 50 basic categories.

**Table 1.** Distribution of Rural Tourism Resources Development in 21 Prefecture-level Cities of Guangdong Province.

| City | Rural Industry Integration | Rural Settlement Architecture | Rural Historic Sites | Rural Folk Culture | Rural Nature Category | Total |
|---|---|---|---|---|---|---|
| Guangzhou | 52 | 49 | 5 | 31 | 15 | 152 |
| Shenzhen | 8 | 7 | 3 | 6 | 7 | 31 |
| Foshan | 25 | 61 | 9 | 33 | 25 | 153 |
| Dongguan | 2 | 3 | 0 | 3 | 1 | 9 |
| Zhongshan | 12 | 31 | 10 | 17 | 15 | 85 |
| Zhuhai | 21 | 22 | 4 | 15 | 14 | 76 |
| Jiangmen | 6 | 23 | 2 | 11 | 3 | 45 |
| Zhaoqing | 20 | 48 | 1 | 40 | 28 | 137 |
| Huizhou | 104 | 120 | 39 | 125 | 91 | 479 |
| Shantou | 28 | 88 | 26 | 38 | 53 | 233 |
| Chaozhou | 5 | 35 | 8 | 24 | 14 | 86 |
| Jieyang | 80 | 134 | 59 | 111 | 125 | 509 |
| Shanwei | 13 | 20 | 9 | 15 | 20 | 77 |
| Zhanjiang | 56 | 19 | 20 | 52 | 68 | 215 |
| Maoming | 5 | 1 | 0 | 0 | 6 | 12 |
| Yangjiang | 9 | 9 | 5 | 10 | 14 | 47 |
| Yunfu | 102 | 109 | 35 | 96 | 120 | 462 |
| Shaoguan | 124 | 118 | 74 | 156 | 116 | 588 |
| Qingyuan | 57 | 72 | 19 | 84 | 70 | 302 |
| Meizhou | 111 | 206 | 59 | 131 | 108 | 615 |
| Heyuan | 80 | 84 | 37 | 75 | 81 | 357 |
| Total | 920 | 1259 | 424 | 1073 | 994 | 4670 |

## 3. Results

This section is divided by subheadings and provides a concise and precise description of the experimental results, their interpretation, and the experimental conclusions that can be drawn.

### 3.1. Pattern of Rural Tourism Resources

Using ArcGIS 10.3, the nearest point index $r_I \approx 0.17$ for rural tourism resources in Guangdong was measured, according to which its spatial distribution was assessed to have clustering characteristics. In addition, the nearest point index $r_I \approx 0.28$ for rural industry integration, $r_I \approx 0.24$ for rural settlement architecture, $r_I \approx 0.37$ for rural historical sites, $r_I \approx 0.12$ for rural folk culture, and $r_I \approx 0.28$ for rural nature category, all showed cohesive distribution.

The geographical concentration index $G \approx 29.06$ was measured by the geographical concentration index formula. If the rural tourism resources in Guangdong are evenly distributed in each prefecture-level city in the province, the geographical concentration index should be $\hat{G} \approx 21.82$, in which $2G > \hat{G}$ indicates that the distribution of rural tourism resources is more concentrated in the municipal scale of Guangdong Province. The $G$ values of all types of rural tourism resources are greater than $\hat{G}$, indicating that their distribution is more concentrated.

According to the calculated formula, the Gini coefficient of Guangdong's rural tourism resources $S = 0.481$ and the uniformity is 0.519. As shown in Figure 1, there is an obvious concave trend, indicating that Guangdong's resources are unbalanced. As a result, rural tourism resources are unevenly distributed among prefecture-level cities throughout the province and are mainly concentrated in Yunfu, Huizhou, Jieyang, Shaoguan, and Meizhou. These five cities account for 56.80% of the total rural tourism resources in the province.

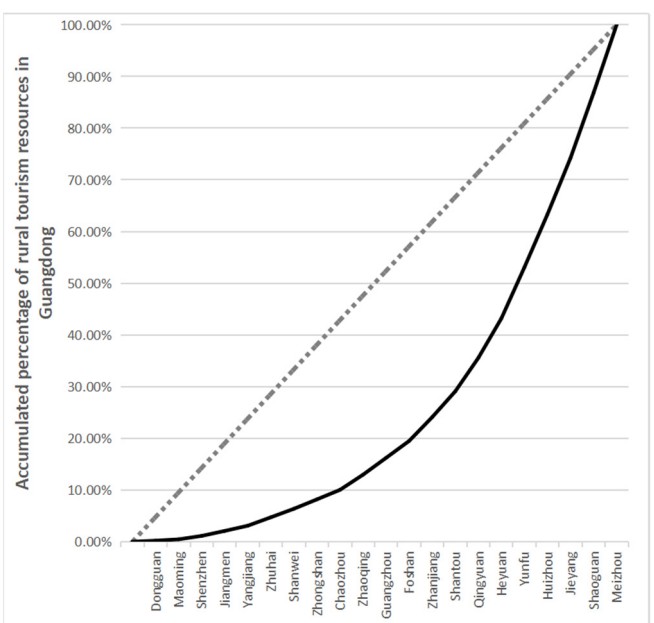

**Figure 1.** The Gini index of the distribution of rural tourism resources in Guangdong Province.

According to the grid-dimensional analysis formula, the probability $P_{ij}$ of Guangdong's rural tourism resources and the value of information quantity $I(r)$ under different grid quantities can be obtained, as shown in Table 2.

**Table 2.** Network dimension measurement data of Guangdong's rural tourism resources.

| K | 2 | 3 | 4 | 5 | 6 | 7 | 8 | 9 | 10 |
|---|---|---|---|---|---|---|---|---|---|
| N(r) | 4 | 8 | 12 | 17 | 25 | 31 | 38 | 47 | 55 |
| I(r) | 1.037 | 1.709 | 2.172 | 2.520 | 2.780 | 3.059 | 3.303 | 3.412 | 3.652 |

Based on (N(r), K) and (I(r), K), the values are plotted on a double logarithmic scatter plot and fitted to a regression to obtain the capacity and information dimension of rural tourism resources in Guangdong Province, as shown in Figure 2. In particular, Figure 2a shows that the coefficient of determination is 0.999, and the spatial distribution of rural tourism resources has a significant scale-free interval with a capacity dimension of 1.6306, which is between 1 and 2 based on the grid dimension. This indicates the uneven spatial distribution of rural tourism resources within Guangdong, suggesting that rural tourism resources are distributed along geographical zones [28]. Meanwhile, Figure 2b shows that the information dimension value is 0.7405 (determination coefficient 0.9649), which is less than the capacity dimension 1.6306. That they are quite different from each other indicates that the spatial distribution of rural tourism resources has a probable unequal posture, complex system fractal structure, grid dimension on the difference of significance, and a strong and significant fractal grid between the differences. The reason for this is that rural tourism resources may be influenced by geographical location and natural climatic factors [31]. Furthermore, rural tourism resources are mainly related to folk culture and settlement architecture, which show a clustered distribution along the Tropic of Cancer and a gradual decrease on both sides. Due to the special geographical location and typical subtropical monsoon climate in the Tropic of Cancer, unique astronomical wonders, cultural customs, and ecological phenomena are formed. Thus, the architectural style, folk legends, and residents' lifestyles have cultural differences from other regions. Finally, they become regional characteristic resources.

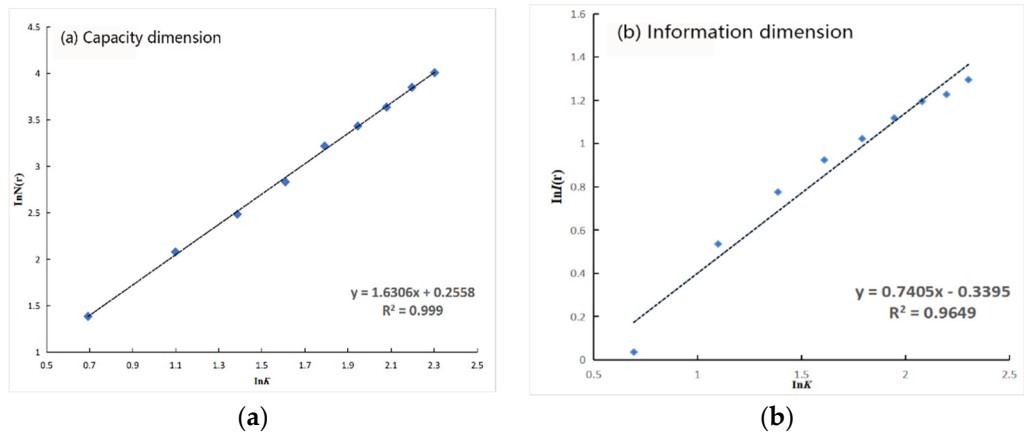

**Figure 2.** Double logarithmic scatter plots of the grid dimension of rural tourism resources in Guangdong; (**a**) Capacity dimension, (**b**) Information dimension.

As shown in Table 3, the Moran's I values of rural industry integration, rural settlement architecture, rural historic sites, rural folk culture, rural nature category, and total tourism resources are 0.024, 0.126, 0.145, 0.081, 0.073, and 0.090, respectively, all of which are greater than 0, thus showing a positively correlated clustering distribution. Rural settlement architecture, rural historic sites, rural folk culture and total tourism resources show a significant edge of spatial correlation. Furthermore, the Z-values are all greater than the upper limit of the uniformly distributed interval of 1.65, indicating an overall spatial agglomeration of rural tourism resources in Guangdong Province, as well as a spatial agglomeration of rural settlement architecture, historical sites, and folk culture.

**Table 3.** Moran's I estimated value statistics of various types of rural tourism resources.

| | Rural Industry Integration | Rural Settlement Architecture | Rural Historic Sites | Rural Folk Culture | Rural Nature Category | Total |
|---|---|---|---|---|---|---|
| Moran's I index | 0.024 | 0.126 | 0.145 | 0.081 | 0.073 | 0.090 |
| Expected index | −0.048 | −0.048 | −0.048 | −0.048 | −0.048 | −0.048 |
| Variance | 0.006 | 0.005 | 0.005 | 0.005 | 0.006 | 0.006 |
| z-score | 0.963 | 2.413 | 2.643 | 1.747 | 1.614 | 1.851 |
| p-value | 0.336 | 0.016 | 0.008 | 0.081 | 0.107 | 0.064 |

The local correlation index was then used to analyze the cold and hotspots (Figure 3). The results show that the cold and hotspots of Guangdong's rural tourism resources have an overall "northeast-southwest" distribution pattern. The hotspot areas are mainly in Heyuan, Yunfu, Shaoguan, and Zhanjiang, while the cold spots areas are mainly in Shanwei, Chaozhou, Zhanjiang, and Zhongshan. Different types of rural tourism resources show the following distribution of high hotspots: rural industrial integration and rural folk culture are mainly in Shaoguan and Heyuan, and rural settlement architecture, rural historical sites, and rural natural categories are mainly in Heyuan.

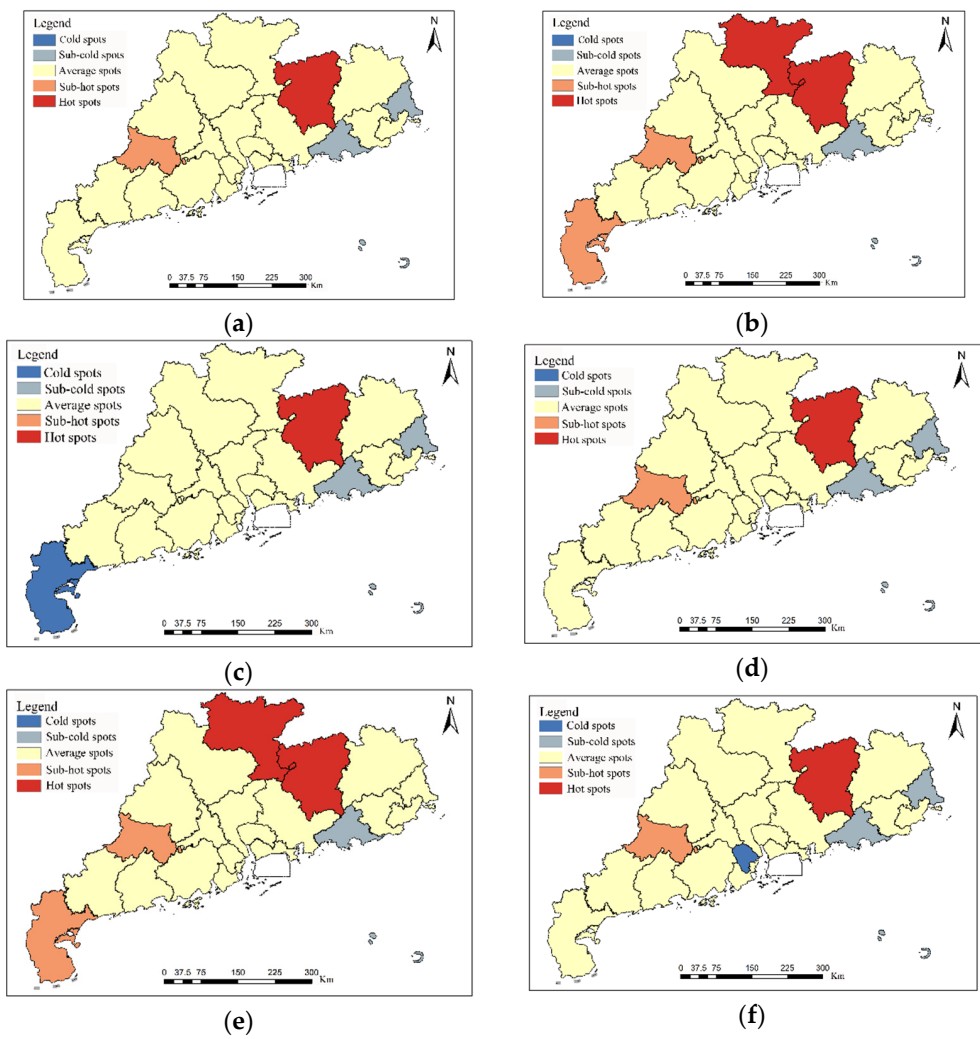

**Figure 3.** Hot spot maps of the spatial distributions of rural tourism resources in Guangdong. (**a**) Total rural tourism resources. (**b**) Rural industry integration. (**c**) Rural settlement architecture. (**d**) Rural historic sites. (**e**) Rural folk culture. (**f**) Rural nature category.

Using ArcGIS 10.3 to calculate the kernel density, the kernel density distribution maps of various types of rural tourism resources in Guangdong Province are obtained, as shown in Figure 4. The rural tourism resources in Guangdong Province generally show a spatial pattern of "three cores, one belt and five poles", and the aggregation characteristics are obvious. Among them, the "three cores" represent three core gathering areas: the area with the highest nuclear density, which is composed of some districts and counties in Meizhou, Jieyang, Chaozhou, and Yunfu. The "Belt" refers to a W-shaped strip distribution pattern formed by connecting multiple low-density areas in the central region of Guangdong Province, mainly involving Yunfu, Foshan, Zhongshan, Huizhou, and Heyuan. The "five poles" refer to five "growth poles", which are medium and low-density radiation areas located in some districts and counties of Shaoguan, Meizhou, Jieyang, Chaozhou, Huizhou, and Yunfu. On the basis of the overall agglomeration pattern, the rural industrial integration class and the rural folk culture class have relatively concentrated point areas in Shaoguan, as well as some districts and counties in Huizhou and Zhanjiang. Rural settlement architecture, rural historical heritage, and rural natural landscape mainly present a spatial aggregation pattern with multiple growth poles.

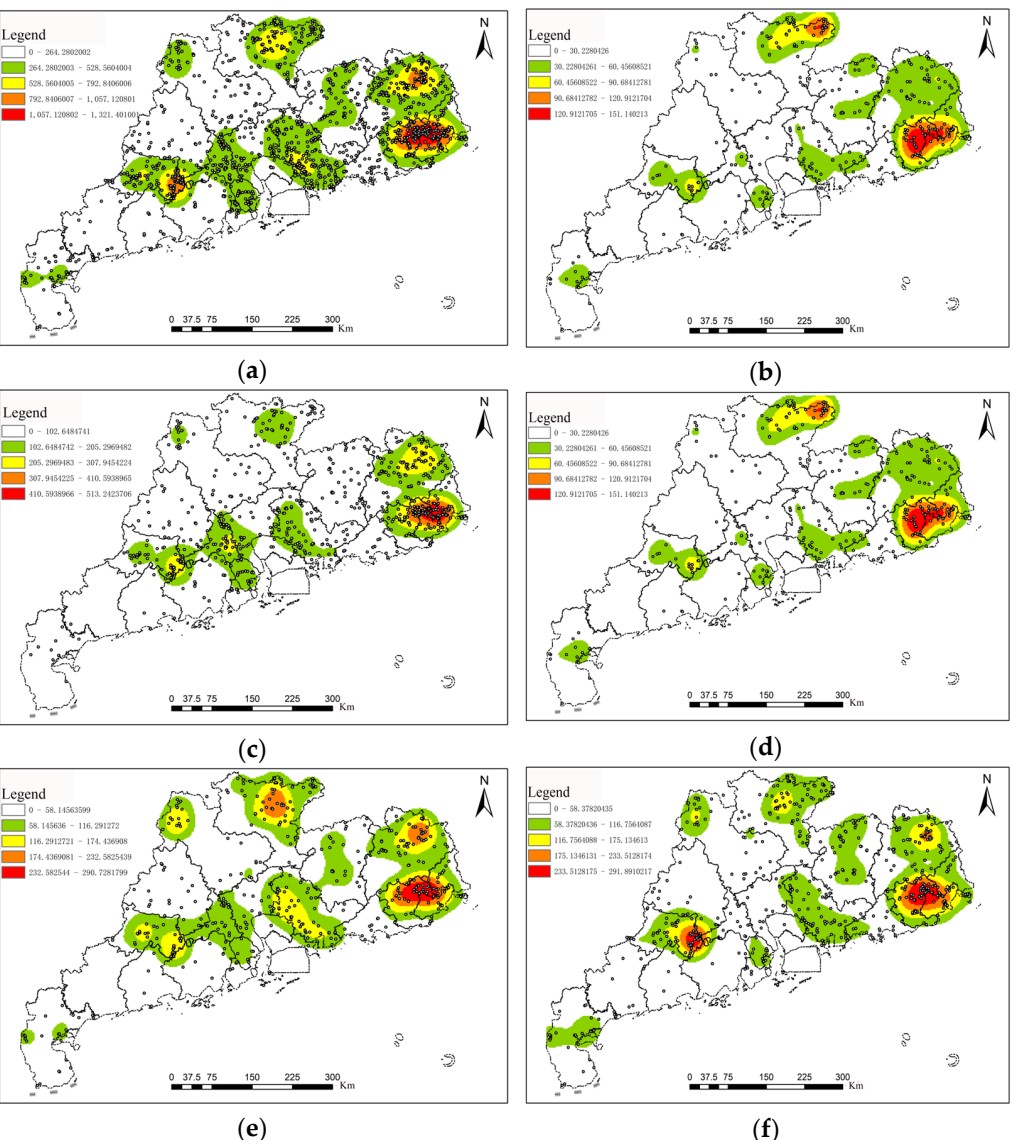

**Figure 4.** Spatial density distribution maps of rural tourism resources in Guangdong. (**a**) Total rural tourism resources. (**b**) Rural industry integration. (**c**) Rural settlement architecture. (**d**) Rural historic sites. (**e**) Rural folk culture. (**f**) Rural nature category.

The standard deviation ellipse method was used to create a more visual representation of the discrete characteristics of the distribution space of each type of rural tourism resources in Guangdong in the latitude and longitude directions. As shown in Figure 5, the first level of standard deviation is used to include the center of mass of about 68% of the total number of input elements. The standard deviation ellipse covers two places: the Pearl River Delta Plain, which is the most densely populated urban and economically developed region in Guangdong Province, and the Chaoshan Plain, which has a warm and temperate climate, fertile land, excellent farming conditions and developed agriculture. They are located in the south-central part and the east coast of Guangdong province. The long axis of the standard deviation ellipse extends from the southwest to the northeast, and the distribution direction of Guangdong's rural tourism resources is mainly in the "southwest-northeast" direction, with an angle of 72.34° and a flatness of 0.54. The directional distribution characteristics are quite obvious, and the center of the ellipse is located in Longmen County, Huizhou City. In addition, all other types of rural tourism resources are spatially distributed in the "southwest-northeast" direction, and the differences are not significant.

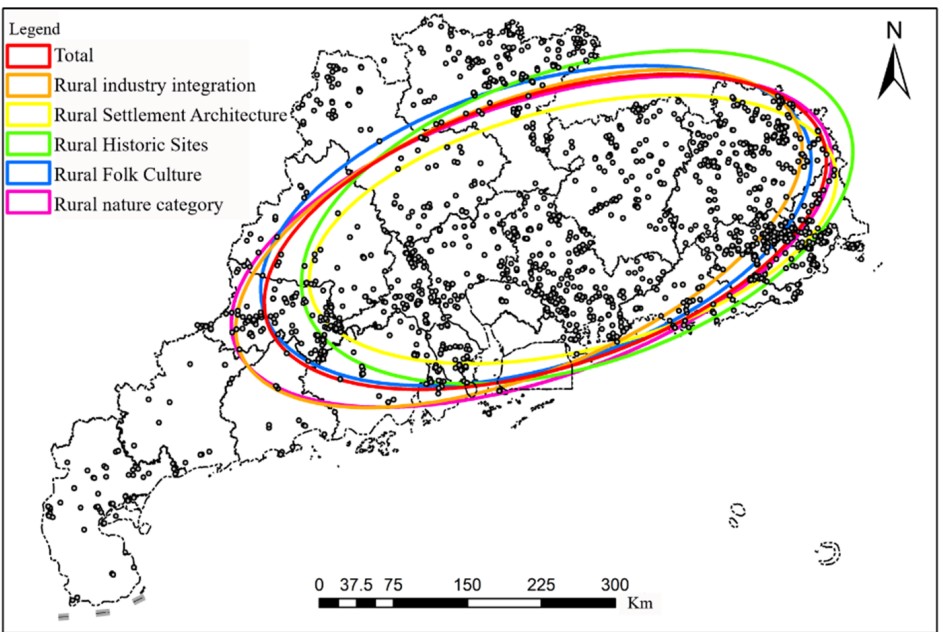

**Figure 5.** Ellipse analysis of the standard deviation of rural tourism resources in Guangdong.

### 3.2. Factors Influencing the Spatial Distribution of Rural Tourism Resource

On the basis of fully understanding the spatial distribution law of rural tourism resources in Guangdong Province, combined with literature review and rural tourism development, based on the availability of data, 12 indicators were selected, including total output values of agriculture, forestry, animal husbandry, and fishery; the number of A-level tourist attractions; and the mileage of opening to traffic. These constitute the five dimensions of agricultural resource endowment, tourism resource endowment, traffic location, tourist market, and social economy. The evaluation results are divided were five categories using the natural segmentation method (Table 4).

**Table 4.** Detection results of the influencing factors of rural tourism resources in Guangdong Province.

| Influencing Factor | Detection Index | q-Statistic |
|---|---|---|
| Agricultural resources endowment | Gross output value of agriculture, forestry, animal husbandry and fishery/billion yuan | 0.502 * |
| | Sown area of major crops/acre | 0.460 * |
| | Total production of major crops/ton | 0.316 * |
| | Number of people engaged in the primary sector/10,000 | 0.256 * |
| Tourism resources endowment | Number of A Grade Tourist Attractions/pc | 0.396 * |
| | Number of 3A Grade Tourist Attractions/pc | 0.296 * |
| Transportation Factor | Miles open/km | 0.333 * |
| | Road density/per km$^2$ | 0.185 |
| Tourism market | Number of resident population/10,000 | 0.125 |
| | Percentage of urban population/% | 0.313 * |
| Socio-economic Factor | Gross regional product/billion yuan | 0.237 |
| | Disposable income per inhabitant/$ | 0.242 |

Note: * represents *p*-values passing the 0.1 significance test.

To explore the magnitude of the explanatory powers of different types of influence factors on the spatial distribution of rural tourism resources, the corresponding probe factor q values of the five dimensions of influence factors were added. The order of descending impact on the spatial distribution of rural tourism resources in Guangdong is as follows: agricultural resource endowment > tourism resources endowment > transportation factor > socioeconomic factors > tourism market. Among them, agricultural resource endowment is greater than 1, which means it is a strong influencing factor. Meanwhile, the explanatory powers of tourism resource endowment and transportation location range from 0.5 to 1.0, which represent a moderate influencing factor. The q values of guest market and socioeconomic factors are both less than 0.5, indicating that they are relatively weak influencing factors. Therefore, from the results, it can be concluded that the factors that have greater influence on spatial pattern distribution of rural tourism resources are agricultural resource endowment, tourism resource endowment abundance, and transportation location factors, while the distribution is relatively less influenced by economic and social environment factors of the source market. The details of the results are as follows:

(1) Regional agricultural development and scale can be reflected to a certain extent by the total output values of agriculture, forestry, animal husbandry, and fishery; the total sown area of major crops; the total output of major crops; and the number of people engaged in the primary industry, which are important indicators to assess the capacity of agricultural development [14]. The stable production of agriculture, forestry, animal husbandry and fishery is the basis for other auxiliary economic activities, which plays an important role in stabilizing economic and social development. Farming, cultivation, sheep herding, horse racing, boat driving, fishing, and other agricultural activities and folklore resources with customs and folklore characteristics form unique rural tourism resources and increase the attractiveness of rural tourism.

Guangdong Province is located in the south of China, with a large number of natural villages, abundant rainfall, and the highest production of fruit, livestock, vegetables, aquatic products, and other agricultural products compared to other provinces in China. The province has numerous mountains, vast tracts of land, and a relatively small population, all of which helped establish the tradition of careful farming that is known throughout the world. The rich characteristic agricultural resources and various high-quality agricultural products helped lay a solid foundation for the coordinated development of agricultural industrialization and rural tourism [27]. Agricultural resource endowment is an important basis for the development of rural tourism, and rural tourism leads to modern agriculture diversification [17].

(2)  Tourism resource endowment provides important support for the development of rural tourism. The richness of tourism resources and resource quality is closely linked to a region's tourism attractiveness and are necessary conditions for the development of rural tourism resources [37]. The performance of China's tourism resource endowment is assessed based on different grade forms of tourist attractions. In this system, A-class scenic spots are the main tourist attractions that serve as resource base for tourism development. Geographic detector analysis found that the tourism resource endowment index has the greatest influence and is, in fact, regarded as a strong influence on the spatial distribution of rural tourism resources. As of March 2021, there are 497 A-class scenic spots in Guangdong Province, including 15 A-class, 187 4A-class, and 281 3A-class scenic spots. The infrastructure and reception facilities of these tourist attractions are relatively perfect, and the good brand effect has a certain radiation effect on the development of surrounding rural tourism resources [21]. The symbolic products formed in the regions with high endowment of tourism resources also have high popularity and appeal in the tourism industry. As they pay more attention to the quality of tourism products, they are able to design special tourist routes, resulting in greater market competitiveness. Moreover, higher-level tourist attractions can provide better external conditions for the development of surrounding rural tourism resources, while the development of rural tourism resources can promote the development of tourist attractions [20].

(3)  Transport accessibility influences the attractiveness of rural tourism development [20]. Many of the characteristics of landscape tourism villages and towns along land, waterways, and other important transportation routes rely on superior transportation location to achieve long-term development. Indeed, underdeveloped transportation location conditions may hamper regional economic development, especially because rural tourism resources are immovable, and rural tourism mainly depends on land-based travel. Land transportation provides an important guarantee for the development of rural tourism, and the breadth and depth of spatial economic ties are affected, to some extent, by the degree of development of transportation [14]. The psychological distance and spatial distance from rural resources will also affect tourists' decisions and have an important impact on regional population, commodity trade, logistics, and transportation [38]. Areas with excellent transportation conditions can attract all kinds of social resources, which means transportation location is an important indicator affecting the spatial distribution of rural tourism resources. Furthermore, the development of rural tourism resources should consider the originality and accessibility of rural resources, such as the construction of rural tourism forest roads and slow-moving greenways.

(4)  The tourism source market is closely linked to the rate of tourism demand. The rural tourism market is mainly distributed in urban areas and towns. The population sizes of the local and surrounding areas are a key factor to consider in tapping the rural tourism market [39]. With the improvement of the economic level and the expectation of retreating from the hustle and bustle of urban life, urban residents who are close to rural tourism areas are the main stakeholders in rural tourism. The location of rural tourism resources adjacent to the tourist market ensures a strong source of guaranteed visitors. In addition, population aids in the development of rural tourism as it provides a large number of funds and sufficient human resources [14]. Given that population is the key external driving force that drives the development of rural tourism, therefore, the source market is the main factor affecting the spatial distribution of rural tourism resources.

(5)  The level of economic development is another driving factor in the development of rural tourism [40]. Regional economic level has improved, along with the accumulation of residents' disposable income, urban residents' consumption desire, and their purchasing power in relation to rural tourism products. Due to the increase of potential consumers in the rural tourism market, tourism capital has been continuously

invested, and the locations of rural tourism resources have gradually developed into a tourism destination, thus improving the ability of tourism destination management activities and product competitiveness [27]. In addition, the increased level of economic development provides support for the upgrading of software and hardware facilities as well as environmental improvements for rural tourism, which in turn, promote its prosperous development. In fact, in 2021, Guangdong's agricultural and rural economic data showed that agricultural production was bountiful, rural leisure tourism business income resumed growth, and new agricultural business entities expanded. The degree of economic development provides sufficient funds and complete infrastructure for the protection and development of rural tourism resources. Furthermore, rural tourism has become a new economic growth point in some local areas, injecting new vitality into the preservation of rural settlement buildings and the inheritance of folk culture.

In summary, by analyzing the factors influencing the spatial distribution of rural tourism resources in Guangdong Province, it is concluded that the spatial distribution characteristics of such resources are influenced by a combination of multiple factors. In particular, agricultural and tourism resource endowments form the basic pattern of "more in the east and less in the southwest," transportation location has a guiding role in the spatial distribution of rural resources, source markets influence the scale of rural resource distribution to a certain extent, and social economy provides a guarantee for the development of rural resources. Overall, the spatial distribution of rural tourism resources is not determined by a single factor, and the leading factors in different regions are different. Therefore, in the spatial development of rural tourism resources, the multiple roles of all the relevant factors should be considered.

## 4. Discussion

As the world economy continues to develop, tourism is also growing rapidly. In recent years, many countries have relied on local ecological, agricultural, and local cultural resources to develop special products and routes, constantly enhancing the appeal of rural tourism. The countryside is becoming an increasingly popular tourist space for tourists from all over the world, as they long to return to nature and experience the local customs and traditions. Thus, when developing rural tourism in developing countries, spatial planning and integration of resources should be optimized to promote regional linkage and clustering for synergistic development. Linkage and collaboration among resource areas and highly rated tourist attractions should be actively promoted, which in turn, will drive the construction of surrounding rural tourism areas. The innovative implementation of rural tourism branding business strategy, in the form of "tourism +," signals the formation of the regional characteristics of the new business model. In addition, this study finds that folk culture in rural tourism resources is the focus of rural tourism development, which is in line with the findings of Petrzelka et al., after surveying the attitudes of the residents of rural tourism sites [41], and with the view of MacDonald and Jolliffe [42]. In the information age, the use of information technology to explore rural resources innovatively, to maintain the original flavor of the countryside in the development process, and to use modern information technology to bring rural culture to life has become highly prevalent. In post-COVID-19 tourism, it is important to establish the rural tourism information service system to promote tourism's rapid recovery [43].

It is necessary to point out that the potential for development exists for some lesser-known rural tourism sites, as cold spot areas have less human damage and disturbance, and for this reason, they have an ecological advantage. This natural and well-preserved countryside is an important condition for the development of wellness tourism. In addition, camping tourism has become a new trend in popular recreation, and some cold spot areas are the best choice for camping because of their vast terrain and good ecological environment. Therefore, in the cold spot areas, we should investigate further into the characteristics of local culture and tourism resources potential. Based on ecological and rural cultural

characteristics, we are able to build several rural camping bases, and to create an important ecological recreation base, to inject new momentum for rural tourism development.

## 5. Conclusions

Based on the analysis of the spatial distribution characteristics of rural tourism resources in Guangdong Province, development paths are proposed according to the resource characteristics of the region. The path of rural tourism development should follow scientific and reasonable planning and adhere to the guidance of scientific theories, so that rural tourism can realize rural revitalization in a healthy and orderly manner [1]. Furthermore, authorities should look into clarifying the market demand of the guest source, fully exploring the local resources of the countryside while considering the ecological, cultural, and social objectives, and not emphasizing market interests over ecology. Moreover, to avoid homogeneous and disorderly competition, the government should enhance the participation of the rural community. This paper proposes the following development paths:

(1) The rural industrial integration resources must include rural folk culture, which is the highlight of developing rural tourism [15,44]. Therefore, in the process of rural tourism development, the government should adhere to the local culture as the basis, grasp the development orientation of local cultural roots, unearth the connotations of rural culture, and integrate these with potential or existing products to avoid homogenization and simplification [45]. For villages with profound cultural heritages, it is necessary to integrate the production, ecological, living, and cultural resources of the countryside; transform the resources into capital to promote rural development; and form a positive synthesis among them [20]. For example, Japan's Shirakawa countryside has several thatched-roof farmhouses with a history of more than 250 years. Furthermore, the area is known for its traditional art of dyeing and weaving and handmade soba noodles, which comprise its local characteristics. Thus, authorities focused on the mining, dyeing, and weaving cultures, Japanese food culture, and the creation of a cultural experiential rural tourism industry [46]. In another example, favorable climate conditions, environmental potential, natural, historical and cultural tourism zones in the East Azerbaijan Province of Iran are the most important strengths [47]. When developing rural resources of folk culture category, the historical authenticity should be displayed for tourists [48]. At the same time, authorities should also strengthen the inheritance and protection of folklore and folkways, develop them scientifically and protectively, and release the driving role of rural folk culture resources in promoting the deep renewal of rural industrial integration.

(2) It is necessary to view rural tourism development from the perspective of coordinated regional development. The rural historic site resources in Guangdong Province are advantageous in that they can promote the development of rural tourism. Related to this, authorities must integrate rural historic sites, including historical allusions, events, and buildings, in order to design historically themed tourist routes. Guided by the needs of tourists, historical sites are places where tourists can "talk directly" with history to address tourists' sense of cultural acquisition [17]. These can be combined with holographic projection and VR to reproduce local life scenes and further enhance the tourist experience. For example, Taos Pueblo, known as the oldest Native American village, is a historical landmark of world cultural heritage, authenticity as a critical factor in rural village experience [49]. During development, it is possible to build a sightseeing route with the theme of Taos history and culture around the collection of houses on both sides of the Taos River, showing tourists the historical charm of old traditional buildings and enabling them to experience the leisurely life in the countryside far away from the metropolis [50].

In the development of various rural historical relic types of resources, it is also necessary to balance the relationship between the protection of historical relics and tourism development. The government should take overall protection measures for historical and

cultural landscapes, retain the authenticity of historical relics, promote the integration of historical culture and tourism, and help boost the revitalization of rural industries [1].

(3)　As for rural areas with attractive natural ecological landscapes and wellness cultures, the government should make rational use of ecological resources, such as forests, bays, and mountains in rural areas, integrate health care projects into the natural environment, protect the regional ecological environment, improve the construction of the tourism industry, and build a comfortable health care tourism base. The authorities should also rely on local resources to extend the forest, fog, solar, ecological, and hot spring baths, as well as other product lines according to local conditions, to create functional health tourism destinations [51]. The environment and resources appeal to tourists, but retaining tourists is still dependent on the culture [52]. To develop a recreation and tourism destination, we should promote the integration of recreation and cultural industries, focus on cultural and experiential aspects, extend the recreation industry chain, and create recreation cultural brand advantages [53]. For example, the organic combination of rural special diet and recreation tourism, which highlights the health and therapeutic aspects of rural food culture, along with local fresh agricultural products, can help develop special diet recreation projects. The Austrian town village of Hallstatt, which has the world's oldest salt pits, has natural salt caves that are valuable for promoting health and boosting the immune system, in addition to serving as sightseeing tour destinations [54]. Such rural resources with health functions should focus on the development of recreation tourism based on special healing services and natural ecological landscapes. Health tourism relies on rural high-quality natural environment and simple local culture, thus breaking through the business model of the original agriculture and traditional tourism. More importantly, it also promotes the adjustment of rural industrial structure and the subsequent development of rural related industries.

Through the analysis of the spatial distribution characteristics of rural tourism resources in Guangdong Province, China, this study reveals the influencing factors of the distribution of rural tourism resources. Practical significance is provided for the scientific and reasonable layout, and the appropriate and sustainable development of rural tourism resources. However, in future research, this study can still be further explored from the following perspectives: (1) case studies can be compared in a more detailed manner with other regions in China or other countries; (2) institutional protection should be strengthened and a system of spatial management of rural tourism resources should be established in conjunction with the regional strategic layout; (3) the protection of ethnic culture, traditional culture and ecological environment should be strengthened, and a digital resource base for rural tourism should be established using information technology; (4) supporting talents, technology, capital and various infrastructure should be injected in conjunction with the influencing factors of the spatial distribution of rural tourism resources; (5) the dynamic evolution of the distribution patterns of different types of rural tourism development resources should be revealed.

**Author Contributions:** Conceptualization, Y.Z.; methodology, Y.Z. and Y.W.; software implementation, Y.Z.; validation, C.L. and Y.W.; investigation, C.L.; data curation, Y.Z. and Y.W.; writing—original draft preparation, Y.Z., C.L. and R.L.; writing—review and editing, Y.Z., C.L., R.L. and M.Z.; visualization, Y.Z. All authors have read and agreed to the published version of the manuscript.

**Funding:** This research was funded by "Evaluation of Sports Tourism Integration and Synergistic Development in the Guangdong-Hong Kong-Macao Greater Bay Area City Cluster, grant number 20BTY054". It was also funded by Shenzhen University 2022 Jutu Teaching Project entitled "Research on the Integration of Sports Tourism Industry Driven by Digital Economy in the Guangdong-Hong Kong-Macao Greater Bay Area, grant number 803-0000311657".

**Data Availability Statement:** The raw data supporting the conclusions of this manuscript can be made available by the authors to qualified researchers.

**Acknowledgments:** We would like to thank the four anonymous reviewers and the editors for their valuable comments and suggestions.

**Conflicts of Interest:** The authors declare no conflict of interest.

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
