# Peer review of "Spatial Differentiation, Influencing Factors, and Development Paths of Rural Tourism Resources in Guangdong Province"

_land, doi:10.3390/land11112046_

Round 1

Reviewer 1 Report

1.This paper has some suggestions with hotspot area in discussion, but how to adjust the less popular places is not mentioned.

 2. P11 Line 381-396 . your paper mentions the influence of different indicators, but does not fully explain the factors that cause such influence? I Suggest supplementary explanations so that readers can fully understand the correctness of your research

 3.  If the government is investing in limited resources, through the research in this article, how do you think it should allocate resources in different regions?

Author Response

Point 1: This paper has some suggestions with hotspot area in discussion, but how to adjust the less popular places is not mentioned.

Response 1: Considering the Reviewer’s suggestion, we have added the development suggestion of cold spot areas to the manuscript on P16 544-554.

Point 2: P11 Line 381-396 . your paper mentions the influence of different indicators, but does not fully explain the factors that cause such influence? I Suggest supplementary explanations so that readers can fully understand the correctness of your research

Response 2: We think this is an excellent suggestion. We have added an explanation of the factors that cause such influence on P12,Line 416-421.

Point 3: If the government is investing in limited resources, through the research in this article, how do you think it should allocate resources in different regions?

Response 3: Thank you for your good questions. Hope the following answers will satisfy you.

  1. For rural areas that are constrained by the level of economic development and have a low level of tourism infrastructure, the government should focus its financial resources on solving the problem of incomplete infrastructure such as public transport, security and communication.
  2. In rural areas that are endowed with tourism resources and are well known, the government should invest funds and launch policies to support the establishment of model villages for rural tourism.
  3. In terms of publicity, support villages to organize various folklore activities, and actively disseminate creative content featuring folklore and culture, so as to establish a unique brand image of rural tourism.
  4. In rural tourism areas with a variety of integrated industries, the government needs to train comprehensive talents to continuously develop and upgrade products to get rid of homogeneity.

Reviewer 2 Report

The reviewed text is very interesting, richly documented, illustrated, and statistically studied. BUT

It lacks a description of the study area. The reviewer does not know the administrative division and also the geographical environment. The authors refer to this in the text  for example "the Pearl River Delta Plain and the Chaoshan Plain, mainly including the central and southern regions of Jiangmen, Foshan, Guangzhou, Zhongshan, and Shenzhen, as well as the eastern regions of Jieyang and Shantou."

There is a lack of literature in the introduction and conclusion on the Japanese, Italian, French, Eastern Europe and Pacific examples described. This is the weakest part of the article  

other

The shape of the ellipse refers to the shape of the study area 

recommended 

Wojciechowska,  Jolanta A summary assessment of the Agritourism Experience in Poland, The Journal of Tourism and Cultural Heritage PASOS – Revista de Turismo y Patrimonio Cultural, 2014, vol. 12 (3), Special Issuse, p. 565-579.
JAŻDŻEWSKA, Iwona. The use of centrographic measures in analysing the dispersion of historic factories, villas and palaces in Łódź (Poland). Folia Geographica, 2018, 60.1: 50.

Author Response

Point 1: The reviewed text is very interesting, richly documented, illustrated, and statistically studied. BUT It lacks a description of the study area. The reviewer does not know the administrative division and also the geographical environment. The authors refer to this in the text, for example "the Pearl River Delta Plain and the Chaoshan Plain, mainly including the central and southern regions of Jiangmen, Foshan, Guangzhou, Zhongshan, and Shenzhen, as well as the eastern regions of Jieyang and Shantou."

Response 1: We think this is an excellent suggestion. We have added a description of the study area on P3 Line 123-127, and updated the description of the Pearl River Delta Plain and the Chaoshan Plain on P11 372-377.

Point 2: There is a lack of literature in the introduction and conclusion on the Japanese, Italian, French, Eastern Europe and Pacific examples described. This is the weakest part of the article  

Response 2: We sincerely appreciate the valuable comments. We have checked the literature carefully and added more references on the Japanese, Italian, French, Eastern Europe and Pacific examples described into the INTRODUCTION and CONCLUSION part in the revised manuscript.

Point 3: other

The shape of the ellipse refers to the shape of the study area

recommended

Wojciechowska,  Jolanta A summary assessment of the Agritourism Experience in Poland, The Journal of Tourism and Cultural Heritage PASOS – Revista de Turismo y Patrimonio Cultural, 2014, vol. 12 (3), Special Issuse, p. 565-579.

JAŻDŻEWSKA, Iwona. The use of centrographic measures in analysing the dispersion of historic factories, villas and palaces in Łódź (Poland). Folia Geographica, 2018, 60.1: 50.

Response 3: As suggested by the reviewer, we have added the above references on P2 Line 66-69,P5 Line 218-220.

Reviewer 3 Report

The work addresses a topic of scientific interest and has been presented in a clear and coherent manner. However, I suggest the following revisions:  

1. The last section "5. Discussion" does not seem like a discussion, but rather a contextualization section of the topic with examples from other countries, more typical of the introduction, although it lacks of citations. Therefore, this section should be reconsidered as an introduction or as part of the conclusions, which by the way, does have a discussion. 

2. The legends of figures 3 and 4 are not legible, so I recommend a revision of the format of both figures so that the legends can be read.

Author Response

Point 1: The last section "5. Discussion" does not seem like a discussion, but rather a contextualization section of the topic with examples from other countries, more typical of the introduction, although it lacks of citations. Therefore, this section should be reconsidered as an introduction or as part of the conclusions, which by the way, does have a discussion.

Response 1: We agree with the reviewer’s assessment. Accordingly, throughout the manuscript, we have revised the DISCUSSION, and added some references to support these examples. Then, we move these examples to the INTRODUCTION.

Point 2: The legends of figures 3 and 4 are not legible, so I recommend a revision of the format of both figures so that the legends can be read.

Response 2: Thank you for pointing this out. This has been clarified in the revised version of the manuscript, and Figures 3 and 4 have been included in the revised manuscript as new Figures 3 and 4.

Reviewer 4 Report

The authors have identified an interesting topic for their research that provides some useful insights in order optimize the layout of rural tourism to explore the spatial differentiation of rural tourism resources and their influencing factors as the authors stated, taking more than 4600 rural tourism resources from Guangdong Province as the research object.

The manuscript is written in a clear and assertive manner, the content of the article reflects the title, are accordingly to the subject analyzed, and each topic that are written on abstract are detailed clearly in the manuscript.

The abstract is specific and concise. It contains the motivation, the purpose of the work and its significance, the main results and major conclusions of the study.

Despite the fact that the study are based on some relevant publications/references available for the topic, it will be useful if the authors will increase the number of references, because mainly the Introduction section could provide a literature review in a broader context (the study could be relevant to a broader academic community if the case study would be treated using comparisons with other regions from China or from some other countries in a more detailed manner).

The methodology is coherent and suitable for the study of the identified and exposed problem. The methods section adequately explain and justify the approach taken, and it is replicable (as per journal guidance). The cartographic and graphic basis are clear and help the reading of the manuscript.

We appreciate that the authors highlighted in a convincing manner the findings and the practical of this study and make the references to policy prescriptions that derives from this analysis, but it could be interesting if the findings will be accompanied with more arguments in a broader context in order to allow others to replicate and build on published results (as per journal guidance). 

We consider that it will be useful if the authors will pointed out the main limitations of the study, as well as the future lines/directions of research and the implications for future research.

Bibliographic references do not meet the requirements of the journal (see Reference List and Citations Style Guide for MDPI Journals). 

Finally, I consider there are many aspects of the manuscript that are of value and likely interest to readers, and also the paper could meet the editorial requirements if the authors will respond to the suggestions made.

Author Response

Point 1: Despite the fact that the study are based on some relevant publications/references available for the topic, it will be useful if the authors will increase the number of references, because mainly the Introduction section could provide a literature review in a broader context (the study could be relevant to a broader academic community if the case study would be treated using comparisons with other regions from China or from some other countries in a more detailed manner).

Response 1: We sincerely appreciate the valuable comments. We have checked the literature carefully and added more references on the Japanese, Italian, French examples described into the INTRODUCTION in the revised manuscript. It would have been interesting to compare with other regions from China or from some other countries. However, to ensure clarity of the research topic, and in view of the difficulty of obtaining data now, this section has been added to the future research  in our study.

Point 2: We consider that it will be useful if the authors will pointed out the main limitations of the study, as well as the future lines/directions of research and the implications for future research.

Response 2: As suggested by the reviewer, we have pointed out our study’s main limitations and future research on P17 636-658.

Point 3: Bibliographic references do not meet the requirements of the journal (see Reference List and Citations Style Guide for MDPI Journals).

Response 3: We have corrected it and we also feel great thanks for your point out.

Round 2

Reviewer 4 Report

I thank the authors for their efforts to respond to all the suggestions made.. 

In my opinion it should be accepted,

Therefore, I recommend this paper for further approval procedures and publishing in the Land journal.

I wish good luck to the authors! 

Author Response

We would like to thank the reviewers for their valuable comments and suggestions again. Thanks very much for your kind work and consideration on publication of our paper. 

Thank you and best regards.